# Epidemiological Assessment and Risk Factors for Mortality of Bloodstream Infections by *Candida* sp. and the Impact of the COVID-19 Pandemic Era

**DOI:** 10.3390/jof10040268

**Published:** 2024-04-03

**Authors:** Jordana Machado Araujo, João Nóbrega de Almeida Junior, Marcello Mihailenko Chaves Magri, Silvia Figueiredo Costa, Thaís Guimarães

**Affiliations:** 1Infection Control Department, Hospital das Clínicas, University of São Paulo, São Paulo 05403-900, Brazil; m.jordana01@gmail.com; 2Central Laboratory Division, Hospital das Clínicas, University of São Paulo, São Paulo 05403-900, Brazil; jnaj99@gmail.com; 3Infectious Diseases Department, Hospital das Clínicas, University of São Paulo, São Paulo 05403-900, Brazil; marcello.magri@hc.fm.usp.br (M.M.C.M.); silviacosta@usp.br (S.F.C.)

**Keywords:** *Candida* sp., epidemiology, mortality, COVID-19

## Abstract

Candidemia is one of the healthcare-associated infections that has high mortality. The risk factors that predispose a patient to develop this infection are mostly found in patients of greater severity and COVID-19 contributes to the risk of death. The aim of this study is to evaluate epidemiological characteristics and risk factors for mortality in patients with candidemia before and during the COVID-19 pandemic era. This is a retrospective study conducted at Instituto Central from 2016 to 2020 of patients with candidemia that were evaluated for demographic data, medical history, risk factors, microbiological data, therapeutic measures, complementary exams, device management, and outcome defined by 30-day mortality. A total of 170 episodes were included (58.2% males; mean age of 56 years). The overall incidence density of candidemia per 1000 admissions and per 1000 patient-days was 1.17 and 0.17, respectively, with an increase of 38% in the year 2020. The use of a central venous catheter was the most prevalent (93.5%) condition, followed by the previous use of antibiotics (91.1%). Corticosteroid use ranked seventh (56.4%). *C. albicans* was responsible for 71 (41.7%) of the isolates, followed by *C. tropicalis* and *C. glabrata*, with 34 (20%) isolates each. Echinocandin was prescribed in 60.1% of cases and fluconazole in 37%. Echocardiography resulted in six (5.08%) cases of endocarditis and fundoscopy resulting in two (2.4%) endophthalmitis. The 30-day mortality was 93/170 (54.7%). The risk factors associated with mortality were age (OR 1.03, CI 95% 1.01–1.06), heart disease (OR 7.51, CI 95% 1.48–37.9), hemodialysis (OR 3.68, CI 95% 1.28–10.57), and use of corticosteroids (OR 2.83, CI 95% 1.01–7.92). The COVID-19 pandemic had an impact on the increase incidence of candidemia. The persistently high mortality highlights the need for better management strategies, control of risk factors, and guarantee of adequate treatment.

## 1. Introduction

Candidemia refers to the isolation of *Candida* species in the bloodstream and this infection can occur through exogenous or endogenous routes [1]. Most infections occur endogenously through the disrupted intestinal epithelial barrier or via indwelling catheters, which represent the main entry port for *Candida* into the bloodstream with subsequent hematogenous spread to other organs [2].

The risk factors that predispose a patient to develop a *Candida* infection are mainly present in critically ill patients, such as previous exposure to broad-spectrum antibiotics, extremes of age, abdominal surgery, solid organ or hematopoietic stem cell transplantation, chemotherapy, neutropenia, mucositis, prolonged stay in the intensive care unit (ICU), presence of invasive devices, total parenteral nutrition (TPN), pancreatitis, and malignancy [3,4,5,6].

The study conducted by Kaur et al. analyzed and compared the incidence of bloodstream infections (BSI) due to *Candida* sp. in developed and developing countries (including samples from Brazil). In this study, developed countries had an incidence ranging from 0.06 to 0.46 BSI/1000 patient-days and from 0.21 to 3.8 BSI/1000 admissions [7]. Brazilian data from Kaur et al.’s study showed 0.26 to 0.37 BSI/1000 patient-days and 1.38 to 2.45 BSI/1000 admissions, revealing higher incidences.

With the advent of the COVID-19 pandemic, there has been an increase in the number of patients in intensive care units. A percentage of patients with COVID-19 are severe enough to require follow-up in the ICU, where the risk of candidemia is high. Epidemiological data show that ICU patients with COVID-19 have a higher incidence of candidemia, resulting in higher mortality rates compared to those without COVID-19 [8].

Recent studies, including Brazilian data, show a higher incidence of candidemia during the pandemic period, suggesting that specific risk factors may be involved in this complication [9]. While these patients had fewer candidemia-related risk factors, such as surgeries and neutropenia, they had more acute risk factors linked to COVID-19 care, including immunosuppressive medications and the use of invasive devices. Given the high mortality rate, it is important for healthcare workers to implement an active surveillance and measures to prevent candidemia in patients with COVID-19 [10].

Considering the importance of these infections in Brazil, in terms of incidence and mortality, and also with the advent of the COVID-19 pandemic, we evaluated the epidemiological characteristics, as well as the risk factors for mortality from bloodstream infections by *Candida* sp. before and during COVID-19 pandemic era.

## 2. Methods

### 2.1. Design

This is a retrospective laboratory surveillance study, based on the analysis of data collected from medical records, to determine the epidemiological, clinical, and microbiological characteristics of episodes of bloodstream infections by *Candida* sp. documented at the Instituto Central of Hospital das Clínicas (ICHC) from 1 January 2016 to 31 December 2020.

The ICHC is a tertiary teaching hospital, with 910 beds, 97 of which are for intensive care units. It also has clinical and surgical inpatient units for various specialties, as well as solid organ (kidney and liver) and hematopoietic stem cell transplant services.

The ICHC underwent a change of operation starting from April 2020, with full mobilization to care for patients with COVID-19, gradually increasing the number of beds in the intensive care unit and ward for these patients. During the period from April 2020 to September 2020, the ICHC had 300 intensive care unit beds and 600 ward beds dedicated exclusively to the care of patients diagnosed with COVID-19. After this period, the hospital underwent demobilization and gradually began to care for COVID-19 and non-COVID-19 patients.

### 2.2. Inclusion and Exclusion Criteria

All patients admitted to ICHC who presented, during the study period, at least one positive blood culture for *Candida* sp. were included. Only the first episode of candidemia was considered for the clinical characterization of the cases.

Outpatients, patients hospitalized in neonatology, patients whose positive culture was only from catheter blood, and patients who had already been hospitalized with a diagnosis of candidemia from another service were excluded.

### 2.3. Calculation of Incidence Density

Based on epidemiological studies, we calculated the incidence density of candidemia per 1000 admissions and per 1000 patient-days using the number of candidemias in the period as the numerator and patient-days and admissions in the same period as the denominator [7].

With these calculations, we performed an incidence curve of the occurrence of candidemia over the years at the ICHC.

### 2.4. Data Collection

Data collection was performed retrospectively, based on positive blood cultures for *Candida* sp. provided by the microbiology laboratory. Epidemiological data were collected through the analysis of medical records and entered in an Excel spreadsheet, created for this purpose.

The patients were followed up during their hospitalization, from the identification of candidemia until discharge from the hospital (discharge or death).

A spreadsheet was filled out with information regarding demographic data, medical history, risk factors for candidemia, microbiological data, therapeutic measures and complementary exams, device management, and assessment of clinical outcome (whether discharge or death during hospital stay). For the analysis of risk factors for mortality, we considered mortality within 30 days after the diagnosis of candidemia.

### 2.5. Microbiological Data

The microbiology laboratory, upon receiving a positive blood culture bottle, performs a gram staining on one sample and inoculates the other onto chocolate blood agar medium and incubates it for 24–48 h, during which time most colonies grow, and may continue a little longer in some species. Then, a small portion of a colony is seeded onto the plate and covered with 0.5 microliter of formic acid. Immediately after drying at room temperature, 1 microliter of matrix (CHCA—alpha-cyano-4-hydroxycinnamic acid) is added and then the plate is placed in the MALDI-TOF.

Colonies or ribosomal protein extracts are superimposed by a matrix on an energy-conducting metal plate. After crystallization of the matrix along with the sample, the metal plate is introduced into the mass spectrometer where it is bombarded with brief laser pulses. The desorbed (released) and ionized molecules are accelerated by means of an electric field and passed through a metal vacuum tube (flight tube) until they reach the detector. Ions with a smaller size (lower mass-to-charge ratio) travel faster through the flight tube than those with a larger size. In this way, the ionized molecules of the samples form mass spectra according to their m/z ratio (mass/charge) and to the peaks that indicate the variable amounts of ribosomal proteins. Finally, for the identification of the microorganism, each peak generated by the analysis of MALDI-TOF MS is compared with a reference database.

### 2.6. Statistical Analysis

Data analysis was performed using relative frequency and position and dispersion measurements. For the analysis of risk factors for 30-day mortality and the inferential analysis of qualitative variables, we used the determination of association using Pearson’s Chi-square test (X2) or Fisher’s Exact Test when the assumption to apply X2 was not satisfied. For the analysis of the differences between the means of the quantitative variables, the one-way ANOVA method was used.

Potential factors related to mortality were compared by bivariate analysis and all factors identified by this analysis as significant were submitted to multivariate analysis, performed by the multiple logistic regression model. To assess the effect of prognostic factors on survival, regression models were used that relate survival times and covariates.

The independent variables were expressed through their risk ratio (“odds ratio”—OR) and their respective 95% confidence intervals (CI) were estimated. All significance probabilities presented were bilateral and performed considering a significance level of 0.05 or 5.0%. Statistical calculations were performed using EPI-INFO version 7.2 and IBM SPSS Statistics version 29.0.1.0. This study was approved by the local ethics committee under number 4.443.713.

## 3. Results

During the study period, 248 patients with positive blood cultures for *Candida* sp. were initially included, but 78 patients were excluded. Of these, forty-six were excluded for having blood cultures collected only from the central venous catheter, without paired peripheral blood cultures or with negative peripheral blood cultures, three for being outpatients, nineteen considered admission candidemia (admitted with a diagnosis of candidemia or transferred from another service already undergoing treatment) and ten for being patients from other institutes, thus resulting in 170 episodes of candidemia for analysis.

Of these, 99 (58.2%) occurred in males and 71 (41.8%) in females. The median age of patients was 56 years old (4–86 years old).

The time between hospital admission and the episode of candidemia was very variable, ranging from zero (i.e., patients who had a positive blood culture collected on admission) to 387 days, with a median of 15 days. Sixteen (9.4%) of the patients had a positive blood culture within the first 72 h of admission.

The incidence density (ID) of candidemia per 1000 admissions and per 1000 patient-days during the study period was 1.17/1000 admissions and 0.17/1000 patient-days. The ID of candidemia per 1000 admissions month by month during the study period is shown in Figure 1.

When we compared the ID per year, we found 2.13, 1.06, 0.71, 0.85, and 1.18 candidemias per 1000 admissions, respectively, from 2016 to 2020, representing an increase of 38% in 2020 (COVID-19 pandemic) when compared to previous years.

Table 1 summarizes the characteristics of the analyzed patients, including comorbidities and risk conditions associated with candidemia.

The microbiological analysis of the species showed eight different species of *Candida* sp., the most common being *Candida albicans* (41.7%), followed by *C. tropicalis* and *C. glabrata* in the same proportion (20%). Table 2 shows the frequency of each identified species.

Regarding treatment, one hundred and eighteen patients (69.4%) underwent treatment, forty-seven (27.6%) died before starting any antifungal therapy and five (2.9%) did not undergo treatment, and only one of these evolved to death in 30 days. The reason for non-treatment was not assessed.

The mean time to start treatment from blood culture collection was 3 days, ranging from −8 (i.e., patients who were receiving empiric antifungal treatment) to 19 days.

Of the treated patients, the antifungal of choice as initial therapy was echinocandin (anidulafungin and micafungin) in 71 (60.1%) cases and fluconazole in 44 (37%) of cases. The median treatment time was 14 days, ranging from 1 to 69 days. Among the patients who had a change of regimen, it occurred on average 6 days after the start of treatment.

A total of 159 patients had some central vascular device. Of these, 111 (69.8%) were patients who started antifungal treatment. The rest were excluded from this analysis because they died before starting therapy. The vascular device was removed in 94 (84.6%) cases, and, in 84 (89.3%), the catheter tip was sent for culture, with a positive result in only 33 (39.2%) of these. In only three cases, the species isolated from the catheter tip culture were different from the species isolated from the peripheral blood culture. The time between candidemia and catheter removal ranged from 0 to 21 days (median 2 days), with zero removal with the collection of blood cultures. In 54 (57.4%) cases, the catheter was removed within 72 h of a positive blood culture.

Regarding the collection of control blood cultures to document microbiological negativity, of the 118 patients who underwent some treatment, 108 (91.5%) had a first collection of control blood cultures, which occurred on average 4 days after the collection of the first blood culture with *Candida* sp. The result was positive in 23 (21.2%) of these cases. Among all patients who collected a first control blood culture, 83 (76.8%) underwent a second collection, with 10.8% positivity. The collection of a third control blood culture occurred in 47 (56.7%) of the cases, with 2.1% positivity.

Of the 118 patients undergoing treatment, 73 (61.8%) underwent transthoracic echocardiography (TTECHO) as initial screening for endocarditis, with three positive results for infective endocarditis (IE) and, in 10 (8.4%), transesophageal echocardiography (TEECHO) was used for screening, with a positive case for IE. Among the patients who underwent TTECHO as screening with a negative result for EI, 10 (14.2%) underwent TEECHO, with two positive results for EI. The total incidence of infective endocarditis was six (5.08%) cases.

Fundoscopy was performed in 81 (68.6%) of the 118 patients undergoing treatment with two (2.4%) cases of endophthalmitis. Patients with endophthalmitis did not have a diagnosis of endocarditis. The time between candidemia and fundoscopy ranged from zero to 53 days (median of 5 days).

Among the 170 episodes of candidemia, in 110 (64.7%), the outcome was hospital mortality, but 93 (54.7%) corresponded to mortality within 30 days. The bivariate analysis of risk factors for 30-day mortality is illustrated in Table 3, and Table 4 shows the multivariate analysis of risk factors for 30-day mortality.

## 4. Discussion

We were able to analyze 170 patients with candidemia at our center with a distribution by gender similar to that of the general population and with a mean age of 56 years.

Regarding the incidence density (ID) of candidemia, our study found a rate of 1.17/1000 admissions and 0.17/1000 patient-days. A study carried out at our institution during the period from 1999 to 2006 showed an incidence density of 0.39 to 0.83/1000 patient-days and, later, another study analyzing only the year 2006 showed incidence densities of 1.85/1000 admissions and 0.27/1000 patients-day [11,12]. After 10 years, our study demonstrates a decrease in ID (from 0.83 to 0.17/1000 patient-days and from 1.85 to 1.28/1000 admissions). This reduction was due to an improvement in care with vascular catheters over the years, including training and increased awareness of the healthcare teams and due to the transfer of the oncology service to another hospital.

Comparing our incidence rate with developed and developing countries (including Brazil), our ID is well below [7]. According to Kaur’s work, in the USA, the ID per 1000 patient-days ranges from 0.30 to 0.46/1000 patient-days, with a maximum value of 0.46. In Europe, the ID varies from 0.03 to 0.44/1000 patient-days and in Brazil, this variation is 0.26–0.37 [9].

When analyzing the temporal evolution, there was a downward trend in the incidence of cases until the beginning of 2020, which coincides with the period of the COVID-19 pandemic. Several studies have been published correlating COVID-19 with the increase in cases of candidemia. A Greek study showed that the incidence of candidemia per 100 admissions in the pre-pandemic period was 5.2 and, in the pandemic period, this value was 33.6/100 admissions [13].

In 2020, the ID of candidemia in our series was 1.18/1000 admissions and 0.18/1000 patient-days. A Turkish study showed an ID of 1.6/1000 admissions during seven months of the year 2021 [14]. In Brazil, Pasqualotto et al., compared the incidence of candidemia in patients with and without a diagnosis of COVID-19 in two hospitals, and the result was an increase in ID per 1000 patient-days from 1.43 (hospital 1) and 1.15 (hospital 2) in patients without COVID-19 to 11.83 (hospital 1) and 10.23 (hospital 2) in patients with the infection [15]. Another Brazilian sample compared a pre-pandemic period with a pandemic period in 2020 and found an increase in candidemia ID from 1.54/1000 patient-days to 7.44/1000 patient-days [16].

The previous use of corticosteroids is considered a risk factor for fungal infections, especially in cases of pulmonary aspergillosis. Corticosteroids have also been reported as a risk factor for candidemia [17]. Considering that the administration of corticosteroids is an effective therapeutic measure for the treatment of patients with COVID-19 who require oxygen support, an increase in ID of candidemia in these patients is expected [18].

However, this increase is multifactorial. First, there are questions related to the severity of patients with COVID-19 that constitute risk factors for candidemia, such as a longer ICU stay, the presence of comorbidities, the need for hemodialysis, and the use of broad-spectrum antimicrobials [19,20]. Second, many hospitals, for several months, were dedicated COVID-19 care centers, which led to a reduction in the total number of admissions, reducing the denominators for calculating incidence density [16]. And thirdly, the workload of health professionals and the shortage of human resources can contribute to a lower adherence to infection control practices [21].

It should be noted that our series does not include the years 2021 and 2022, when we still received a large contingent of patients with COVID-19, but the ICHC was no longer dedicated to the exclusive care of these patients.

The analysis of comorbidities shows gastrointestinal diseases as the most prevalent, followed by chronic kidney disease (CKD) and diabetes, while neoplasia is in seventh place in our series. These data vary in the literature, since neoplasia/solid tumors appear more frequently in several studies. A multicenter study conducted in Brazil between 2007 and 2010 showed neoplasia as the most frequent comorbidity (32.1%), followed by gastrointestinal diseases (18.9%). Chronic kidney disease had a low prevalence when compared to our study (6.5% vs. 27.8%) [22]. Likewise, an Argentinian single-center study identified the presence of solid neoplasia as the most frequent comorbidity (23.8%), followed by diabetes, chronic kidney disease, and gastrointestinal diseases [23]. An Italian multicenter study showed a higher prevalence of solid tumors in patients with candidemia admitted to tertiary hospitals, followed by cerebrovascular disease, diabetes and CKD [24].

The main risk factors found were the presence of CVC and the use of antibiotics in the last 30 days. In addition to the comorbidities found, these factors are classically associated with candidemia and this is corroborated in several studies [11,22,25]. The use of corticosteroids appears in 56.4% of cases throughout the period of this study, and, in the pre-COVID-19 period, this value was 53.9% and, in the year 2020, it increased to 68.9%.

Regarding species distribution, we found 41.7% prevalence of *C. albicans* and 58.3% of non-*Candida albicans* species. When we analyzed the non-albicans species we found 20% of *C. glabrata*. When comparing these data with samples from the same institution, we found an increase in the prevalence of this species.

More recently, studies in Brazil reflect this increase in *C. glabrata*. Moretti, et al., in 2013, published a study carried out at the University of Campinas showing 11.2% of this species between 2006 and 2010 [26]. Doi, et al. analyzed 16 public and private centers from 2007 to 2010 and found a prevalence of 10.2% of *C. glabrata* [22]. A study in the city of Uberlândia, between 2009 and 2016, showed a prevalence of 8.5% [27]. Rodrigues, et al. analyzed 22 public hospitals in the state of São Paulo between 2017 and 2018 and showed 9.7% of *C. glabrata* [28]. Agnelli carried out a study in São Paulo comparing two periods, 2010–2011 and 2017–2018, and found 13.3% and 13%, respectively [29].

The increase in *C. glabrata* candidemia is classically correlated with the use of azoles as prophylaxis or as empirical therapy, mainly in ICUs, based on *Candida* scores. In our study, we did not correlate the consumption of fluconazole with the increase in the incidence of this species. The prophylactic use of azoles in the transplant scenario is performed for HSCT during neutropenia; prophylactic use is not routinely used in kidney transplantation and in liver transplantation, only in specific situations, such as fulminant hepatitis, retransplantation, and for a patient on hemodialysis. Thus, we believe that the increased prevalence of *C. glabrata* is not due to inappropriate, or even indiscriminate, use of prophylactic/empirical fluconazole, but is related to epidemiological change, as described in the studies above. The finding that 20% of candidemias are caused by *C. glabrata* implies a change in empirical therapy, which, with this percentage, should not be done with fluconazole. Another important factor to be considered in our study was that the increase in the ID of candidemias in the year 2020 was accompanied by an increase in the ID of *C. glabrata*. This may be related to the increased use of antimicrobials and empirical antifungal use in the context of severity in patients with COVID-19.

Although *Candida parapsilosis* was found in only 12.3% of cases, this species deserves special attention because it is considered an emerging species due to the emergence of azole resistance [30].

We observed that only 69.4% of patients received treatment. Even considering that 27.6% did not have the chance to receive antifungal medication because they died within 3 days of the incident candidemia, there still were 2.9% of patients who did not receive any antifungal treatment and even so survived. It is noteworthy that peripheral blood cultures positive for *Candida* species are rarely interpreted as colonization/contamination and all episodes of candidemia should be viewed with caution and treated, considering the high morbidity and mortality of this infection [22].

Regarding the time to start antifungal treatment, we found an average of 3 days, ranging from −8 (which means patients were using prophylactic antifungals) to 19 days. Regarding this issue, we must point out that the delay in establishing effective therapy increases the risk of death and we need to work to improve stewardship practices and communication between the microbiology laboratory and the attending physician for critical results.

With regard to the initial therapeutic scheme, we observed that, in only 60.1% of the cases, an echinocandin was the initial drug of choice. Several guides and studies recommend echinocandins as a first-line drug to start treatment, with an A-1 level of evidence, based on a systematic review that demonstrated lower mortality in the group that started treatment with echinocandins versus comparator groups [1,2,3,25,31]. Therefore, it is necessary to change prescribing habits and promote continued education for physicians, targeting echinocandins as initial therapy for proven cases of candidemia.

The 30-day mortality rate was 54.7%. Many papers report general mortality ranging from 50–70%. Historical series from our Institute demonstrated, in 1999–2006, a mortality in 14 days of 45–50% [11]. Another study carried out in Latin America showed a 30-day mortality rate of 40.7% [32]. Historical series carried out in Rio de Janeiro, from 1996 to 2016, showed a 30-day mortality rate of 58.9% [33]. In 2021, Husni, et al., published a study whose Brazilian mortality was 63.6%, compared to 20% in the United States and 15.6% in Spain [34].

A study comparing populations from”Brazil versus Spain showed higher mortality at 30 days (51.9% × 31.6%, *p* < 0.001) and at 14 days (35.8% × 20.1%, *p* < 0.001) in our country [35]. Another study by the same author comparing two periods showed a mortality rate of 41.1% (2010–2011) versus 41.2% (2017–2018). These data show that the mortality of this infection in Brazil remains high despite advances in diagnostic and treatment techniques and this may be due to an increase in the complexity of patients, such as the presence of dialytic chronic kidney disease and the use of immunosuppressants or the presence of more than three comorbidities in patients, in addition to suboptimal therapeutic interventions, as reported in these two studies by Agnelli [29,35].

The bivariate analysis showed the following risk factors for 30-day mortality: advanced age, lung disease, heart disease, ICU stay, hemodialysis, use of corticosteroids, use of antibiotics, and mechanical ventilation. We found a higher survival rate in patients with transplantation, surgery, gastrointestinal tract surgery, total parenteral nutrition, candidemia by *C. parapsilosis*, de-escalation to fluconazole, longer time between candidemia and CVC removal, and time between candidemia and start of treatment.

When we performed the multivariate analysis, the following risk factors remained: age, heart disease, hemodialysis, and previous use of corticosteroids. A higher survival rate in patients with transplantation, de-escalation to fluconazole, and longer time between candidemia and CVC removal was observed. Corticosteroids were an independent risk factor for mortality, as demonstrated in several studies and due to the association with COVID-19 [35,36]. It is worth noting that COVID-19 infection itself was not a risk factor for mortality.

Limitations of this study include the impossibility of performing sensitivity tests on the isolated strains. With this data, it would have been possible to provide a more complete microbiological update of candidemias at ICHC in the last five years. In addition, other limitations were the retrospective nature of the study and being a single center, which may not reflect the global epidemiology. Also, as mentioned above, our series does not include the years 2021 and 2022, which were still during the COVID-19 pandemic period.

Considering the high incidence and high mortality of these infections in Brazil and with the advent of the COVID-19 pandemic and, now, in the post-COVID-19 period, the analysis of the epidemiological and microbiological characteristics, including the sensitivity profile and risk factors for mortality of bloodstream infections by *Candida* sp., contributes to a better understanding of the evolution of this infection and, thus, we can try to establish effective prevention and therapeutic measures to reduce the incidence and mortality related to this infection.

## 5. Conclusions

The overall incidence density (ID) of candidemia at ICHC during the study period was 1.17/1000 admissions and 0.17/1000 patient-days, with the COVID-19 pandemic having an impact on the increased incidence of candidemia when compared to previous years.

Gastrointestinal tract disease and the use of a central venous catheter were the most prevalent risk conditions. *Candida albicans* was the most prevalent species (41.7%), followed by *C. tropicalis* and *C. glabrata* in the same proportion (20%), with *C. parapsilosis* found in 12.3% of cases.

A proportion of 69.4% of the patients with candidemia underwent treatment, with echinocandins being the antifungal of choice as initial therapy in 71 (60.1%) of the cases. The incidence of endocarditis and endophthalmitis was 5.0% and 2.4%, respectively.

The 30-day mortality was 54.7%, with the independent risk factors for mortality being age (OR = 1.03; 95% CI 1.01–1.06; *p* = 0.007), heart disease (OR = 7.51; 95% CI 1.48–37.9; *p* = 0.015), need for hemodialysis (OR = 3.68; 95% CI 1.28–10.57; *p* = 0.015), and use of corticosteroids (OR = 2.83; 95% CI 1.01–7.92; *p* = 0.047).

The identification of risk factors contributes to improving management, control, and adequate treatment strategies in order to minimize the unfavorable outcome of these infections.

## Figures and Tables

**Figure 1 jof-10-00268-f001:**
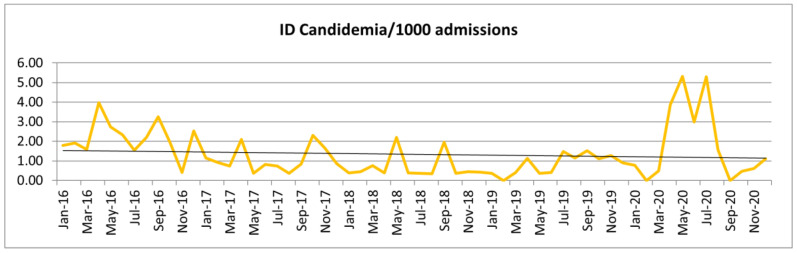
Incidence density (ID) of candidemia per 1000 admissions in the study period.

**Table 1 jof-10-00268-t001:** Baseline and risk conditions in patients with candidemia at ICHC, during the study period (N = 170).

**Baseline Conditions**	**N (%)**
Gastrointestinal Tract Disease	71 (41.7%)
Chronic Renal Failure	38 (22.3%)
Diabetes	44 (25.8%)
Chronic Pulmonary Disease	46 (27.0%)
SARS-CoV-2 Infection	20 (11.7%)
Organ Transplantation	33 (19.4%)
Neoplastic Disease	16 (9.4%)
Chronic Heart Disease	21 (12.3%)
Immunodeficiency	11 (6.4%)
Burn	5 (2.9%)
**Risk Conditions**	**N (%)**
Central Venous Catheter	159 (93.5%)
Previous Use of Antibiotics	155 (91.1%)
ICU Admission	140 (82.3%)
Previous Surgery	86 (50.5%)
Mechanical Ventilation	95 (55.8%)
Hemodialysis	88 (51.7%)
Previous Use of Corticosteroid	96 (56.4%)
Previous Gastrointestinal Surgery	44 (25.8%)
Previous Use of Immunosuppressants	41 (24.1%)
Total Parenteral Nutrition	34 (20.0%)
Neutropenia	8 (4.7%)
Pancreatitis	8 (4.7%)

ICU = intensive care unit.

**Table 2 jof-10-00268-t002:** Frequency of *Candida* sp. species isolated in blood culture at ICHC, during the study period (N = 170).

Species	N (%)
*C. albicans*	71 (41.7%)
*C. tropicalis*	34 (20.0%)
*C. glabrata*	34 (20.0%)
*C. parapsilosis*	21 (12.3%)
*C. dublienensis*	4 (2.3%)
*C. haemulonii*	2 (1.1%)
*C. kefyr*	2 (1.1%)
*C. krusei*	2 (1.1%)

**Table 3 jof-10-00268-t003:** Bivariate analysis of risk factors for 30-day mortality in patients with candidemia included in the study (N = 170).

Variable	Death in 30 Days N = 93 N (%)	Survival in 30 Days N = 77 N (%)	*p*
Demographic Data			
Male sex	60 (64.5%)	39 (50.6%)	0.068
Age (in years)	57.63	47.29	0.0001
Comorbidities			
Diabetes	29 (31.1%)	15 (19.4%)	0.083
Neoplasia	10 (10.7%)	6 (7.7%)	0.511
Immunodeficiency	9 (9.7%)	2 (2.6%)	0.062
Chronic pulmonary disease	33 (35.5%)	13 (16.8%)	0.006
SARS-CoV-2 infection	15 (16.1%)	5 (6.5%)	0.052
Chronic heart disease	17 (18.2%)	4 (5.2%)	0.010
Chronic renal failure	21 (22.5%)	17 (22.0%)	0.937
Gastrointestinal tract disease	33 (35.5%)	38 (49.3%)	0.068
Organ transplantation	13 (13.9%)	20 (25.9%)	0.049
Burn	2 (2.1%)	3 (3.9%)	0.503
Risk Conditions			
Previous surgery	39 (41.9%)	47 (61.0%)	0.013
Previous gastrointestinal surgery	17 (18.3%)	27 (35.0%)	0.013
Total parenteral nutrition	13 (13.9%)	21 (27.2%)	0.031
ICU	85 (91.4%)	55 (71.4%)	0.0006
Hemodialysis	58 (62.4%)	30 (38.9%)	0.0024
Previous use of corticosteroid	62 (66.6%)	34 (44.1%)	0.0033
Pancreatitis	5 (5.4%)	3 (3.9%)	0.651
Neutropenia	5 (5.4%)	3 (3.9%)	0.651
Previous use of immunosuppressants	23 (24.7%)	18 (23.3%)	0.837
Previous use of antibiotics	90 (96.7%)	65 (84.4%)	0.0048
Central venous catheter	87 (93.5%)	72 (93.5%)	0.991
Mechanical ventilation	63 (67.7%)	32 (41.5%)	0.0006
Microbiology			
Time between admission and candidemia (days)	23.03	22.84	0.957
*C. albicans*	39 (41.9%)	32 (41.5%)	0.960
*C. tropicalis*	24 (25.8%)	10 (12.9%)	0.038
*C. glabrata*	18 (19.3%)	16 (20.7%)	0.817
*C. parapsilosis*	5 (5.4%)	16 (20.7%)	0.0024
Other Non-*Candida albicans* species	7 (7.5%)	3 (3.9%)	0.318
Treatment			
Initial treatment with echinocandin	28 (30.1%)	43 (55.9%)	0.901
Initial treatment with azoles	17 (18.2%)	28 (36.3%)	0.833
Time between candidemia and treatment (days)	2.28	3.77	0.009
Step-down to fluconazole	3 (3.2%)	18 (23.8%)	0.010
Device management			
CVC removal	52 (55.9%)	60 (77.9%)	0.208
Time between candidemia and CVC removal (days)	2.47	5.06	0.009
Complications			
Infectious endocarditis	3 (3.2%)	3 (3.9%)	0.814
Endophthalmitis	1 (1.0%)	1 (1.3%)	0.893

**Table 4 jof-10-00268-t004:** Multivariate analysis of risk factors for 30-day mortality in patients with candidemia included in the study.

Variable	OR	CI 95%	*p*
Age (years)	1.03	1.01–1.06	0.007
Chronic heart disease	7.51	1.48–37.9	0.015
Organ transplantation	0.16	0.03–0.73	0.018
Hemodialysis	3.68	1.28–10.57	0.015
Previous use of corticosteroids	2.83	1.01–7.92	0.047
Step-down to fluconazole	0.15	0.03–0.81	0.028
Time between candidemia and CVC removal (days)	0.84	0.73–0.96	0.01

## Data Availability

Data are contained within the article.

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
