# Peer review of "Epidemiological Assessment and Risk Factors for Mortality of Bloodstream Infections by Candida sp. and the Impact of the COVID-19 Pandemic Era"

_jof, 2024, doi:10.3390/jof10040268_

Round 1

Reviewer 1 Report

The authors analyzed relevant data to associate factors that can cause death in patients with candidemia and thus take preventive measures in subsequent cases. Therefore, their work is relevant.

Corrections are proposed in tables 2 and 3, corrections in lines 211, 349 and 354, also some changes in the wording of some paragraphes. Document is attached.

Reviewer 2 Report

Dear authors,

The diagnosis and treatment of invasive candidosis is still a big challenge. Since that diagnosis of invasive candidosis start with consideration of present risk factors, it is very important to evaluate them. However your article must be improved for publication.

My suggestions:

General

Based on the new taxonomy due to their characteristics, yeasts of Candida genera have been assigned to different or newly created genera. For example, the widespread species C. glabrata was added to the Nakaseomyces (N.) genus, and its name is now changed to N. glabrata. Similarly, one of the most prevalent species for human pathology, C. krusei, was reclassified into the Pichia (P.) genus, so now it is referred to as P. kudriavzevii. Species C. guilliermondii and C. kefyr have also been transferred and are now members of the genera Meyerozyma (M. guilliermondii) and Kluyveromyces (K. kefyr) (5). However, the newly determined classification of yeasts poses a new challenge in medical mycology. Adopting and defining isolates according to the new taxonomy could confuse clinicians when interpreting mycological analyses. Many authorities argue against changing the old names of yeasts, now classified into separate genera. You have to point out these facts, but like others you can use old terms.

Besides, you have to improve introduction and discussion, with pointing of diagnostic procedure recommended by competent organizations, criteria necessary for diagnosis and treatment procedure.

Specific

1. It is important to highlight the fact that regarding exogenic infection and horizontal transmission, there are still controversial data. Endogenic infection could be from gastrointestinal tract and from the skin frequently colonized by C. parapsilosis. 

2. Display in order all predispose and risk factors for examples:  chemotherapy, diseases, medical procedures.

3. Since you consider patients with one positive findings of Candida in blood, how did you separate infection from colonization or contamination. Based on which criteria candidaemia was defined

4. Procedure of MALDI-TOF is not clearly explain, it has to be improve

5.  Why patients had different therapy of antifungals, what was the reason for choice of  initial therapy with echinocandin or fluconazole

6. Avoid blanket statements: like  This reduction may be related to an improvement in care with vascular catheters  and there was a change in the Hematological Oncology and HSCT service in 2017 to an other institute during a renovation. After the renovation, the HSCT service returned to the  ICHC, but the Hematological Oncology service remains at another institute in the complex

7. When you discuss and compare results of other surveys you have to highlight if the same diagnostic procedure was used in them. Cultivation or better to say bloodculture as methods has low sensitivity. You can compare results of invasive candidosis, candidemia if they were proved by same methods.

8. Please, avoid to use statistical results of prediction in manners of protection:  How transplantation, de-escalation to fluconazole and longer time between candidemia and  CVC removal could be protective factors.

9. In discussion text from line 257 to 267 has to be in limitation

10. There are not data regarding applied prophylaxis in evaluated cases/patients.

11. Avoid discussing C. glabrata incidence, it is fine to highlight the shift from C. albicans to non-albicans species as causative agent of candidemia and invasive candidosis in general.   

 Your manuscript can be published after major revision based on my suggestions.

Round 2

Reviewer 2 Report

No comments

No comments